

# Distribution, ecological risk assessment and source identification of pollutants in soils of different land-use types in degraded wetlands

Yangyang Han[1,2,3,*], Hongjie Wang[1,2,3,*], Guangming Zhang[4], Shengqi Zhang[1,2,3], Xingchun Liu[1,2,3] and Ling Liu[1,2,3]

[1] School of Eco-Environment, Hebei University, Baoding, China
[2] Institute of Life Science and Green Development, Hebei University, Baoding, China
[3] Hebei Key Laboratory of Close-to-Nature Restoration Technology of Wetlands, Hebei University, Baoding, China
[4] School of Energy and Environmental Engineering, Hebei University of Technology, Tianjin, China
[*] These authors contributed equally to this work.

Corresponding author
Ling Liu, liuling@hbu.edu.cn

## ABSTRACT

**Background**. Urbanization and global warming are generating ecological degradation and land pattern alteration problems in natural wetlands. These changes are greatly affecting the ecological services of wetlands. Therefore, there is an urgent need to explore the relationship between pollutants and land-use type for wetland restoration purposes. Zaozhadian Lake is a freshwater wetland in the North China Plain, which is facing degradation and land-use types changes. An experiment for analyzing soil pollutants was conducted in three land-use types of farmland, lake, and ditch in the Zaozhadian Lake. The aims of this study were to identify the distribution, pollution degree, and sources of pollutants in different land-use types, and to explore the influence of land-use type changes on contamination.

**Methods**. In this study, surface sediments (0–10 cm) of three land types (farmland, lake, and ditch) in Zaozhadian Lake were collected, and heavy metals (Cu, Ni, Zn, Pb, Cd, Cr, Hg), As, total nitrogen (TN), total phosphorus (TP) and organic matter (OM) were determined. Kriging interpolation was used to visualize the pollutants distribution. The pollution degree of TN and TP was evaluated by the Nemerow pollution index. The pollution of heavy metals and As was evaluated by the geological accumulation index ($I_{geo}$) and the potential ecological risk index ($RI$). Then, dual hierarchical clustering analysis and the principal component analysis were performed to further analyze the impact of land type changes on pollutants.

**Results**. The heavy metal contents in the farmland were higher than other areas, while the TN (3.71 ± 1.03 g kg⁻¹) and OM (57.17 ± 15.16 g kg⁻¹) in lake sediments were higher than that in other regions. Farmland, lake, and ditches had low ecological risks, with $RI$ values of 84.21, 71.34, and 50.78, respectively. The primary heavy metal pollutants are Pb, Cu, and Ni. Furthermore, Cu, As, Ni, Pb, and Zn were primarily derived from agriculture pollution, the source of Cd was the industrial pollution, and Cr mainly originated from natural sources. Nutrients primarily came from the decomposition of aquatic animals, plants, and human-related activities. When the lake area was converted into farmland, the heavy metal concentrations in the soils increased

and the TN and OM decreased. Based on the results, this study put forward key strategies including the adjustment of the land-use type and restriction of the entry of pollutants into the wetland ecosystems in the Zaozhadian Lake. More attention should be paid to the impact of land-use type change on pollutants in wetlands.

## INTRODUCTION

Wetlands play an important role in improving water quality, protecting biodiversity, and regulating climate (*Acharya, Maraseni & Cockfield, 2019*). However, heavy metal-like pollutants are continuously flowing into wetlands, along with the global warming events, the destruction of wetlands by humans, and the continuous development of all types of industrial activities (*Lu et al., 2020*). The flowing of pollutants into the wetlands can be affected by hydrodynamic forces, aquatic plants, microorganisms, and other physicochemical factors. These pollutants can be then retained in the soil or accumulated in the plants by precipitation and adsorption (*Jiang et al., 2017*). Excessive pollutants in wetlands can pose potential hazards to natural ecosystems, plants, human health, and other living organisms (*Alahabadi & Malvandi, 2018*). In the past few decades, researchers have conducted numerous studies on soil pollution in wetlands (*Ji et al., 2019a*; *Lu et al., 2018*). Heavy metal contamination has gained wide attention for its toxicity, persistency, and non-biodegradability in the environment (*Puthusseri et al., 2021*). In addition, nitrogen and phosphorus has been a major concern because they are the key limiting nutrients for eutrophication in most aquatic ecosystems (*Conley et al., 2009*).

Recently, the interest in the effects of land pattern conversion on pollutant behavior in soil has increased (*Bai et al., 2010*). According to the Ramsar Convention on Wetlands (*RCW, 2018*), about 35% of the natural wetlands have been lost since 1970 due to urbanization events (*Hu et al., 2017*). In fact, degraded wetlands are often used as agricultural land, and in industrial and urbanization constructions (*Ding et al., 2020*). Many studies have proved that contaminant accumulation can be inextricably linked to land use patterns (*Tian et al., 2019*; *Sun et al., 2019*), and land use conversion can directly or indirectly influence the geochemical position of contaminants (*Jiao et al., 2014*). For instance, *Zheng et al. (2016)* proved that there are fewer heavy metals in the forests of the Yangtze River Estuary compared to the heavy metals present in farmland areas. In addition, *Bai et al. (2010)* showed that there are differences in the distribution of soil pollutants of wetlands, conventional farmlands, and abandoned farmlands in Yilong lake. Other studies analyzed the pollution risk based on land use patterns, such as industrial areas (*Lu et al., 2018*), lakes (*Ji et al., 2019a*), and agricultural lands (*Wang et al., 2019*). However, few studies have focused on the effects of land-use type conversion on heavy metals and nutrients, especially in wetlands. As the population continues to grow and develop, the wetland land patterns

are gradually changing (*Liu et al., 2020*). Therefore, there is a need to analyze the influence of land-use type on contaminants in wetland soils. These finding can provide reference materials in targeted thinking related to sustainable development of wetland.

The Zaozhadian Lake is a comprehensive wetland composed of lake, rivers, and cultivated land, and it has significant characteristics of land-use type change. The studies on the Zaozhadian Lake focus mainly on aspects such as activities of microbial communities (*Sun et al., 2021*), contaminant contents (*Wang et al., 2020*), eutrophication (*Pu et al., 2017*), and ecological risk assessment (*Ji et al., 2019b*). However, these studies did not differentiate contamination based on land-use types. The objectives of this study were to (1) study the spatial heterogeneity of pollutants with different land-use types by using statistical description and geostatistical methods; (2) assess the sediments nitrogen and phosphorus pollution, and evaluate the degree of heavy metal pollution and ecological risk in different regions by the geological accumulation index and the potential ecological risk index; (3) identify the main sources of pollutants; and (4) analyze the influence of land-use change on  contaminants.

## MATERIALS AND METHODS

### Study area

Zaozhadian Lake is located in Hebei Province of North China Plain, which is adjacent to Tianjin and Beijing. Zaozhadian Lake is part of Baiyangdian Lake, which is the largest wetland in the North China. The Zaozhadian Lake plays an important role in reducing the pollution load of the Baiyangdian Lake and adjusting the regional climate. The average annual precipitation in this region is 563.9 mm. Also, the distribution of precipitation along the area during the whole year is extremely unbalanced, with an 80% from June to August (*Wang et al., 2020*; *Yang et al., 2020*). Caohe River, Baohe River, Fuhe River are rivers located upstream of Zaozhadian Lake, and these rivers are also considered to be important water sources of the Baiyangdian Lake. However, domestic sewage and industrial wastewater are usually discharged into these rivers. Recently, the lack of precipitation and intensive human activities have led to the ecological degradation of the study area (*Zhang et al., 2020*), and the land-use types have undergone several changes. For instance, the water area was reduced and the sediments were exposed from 1999 to 2018. At the same time, the cultivated land, ditches, and construction land showed an overall increasing trend (*Lv et al., 2020*). However, the ecological degradation of Zaozhadian Lake has greatly reduced its ability to purify pollutants.

### Sample collection and analysis

Figure 1 shows the 25 sampling sites collected on September 2019 and used in this study. Sample collecting and field experiments were approved by the Major Science Technology Program for Water Pollution Control and Treatment of China (2018ZX07110). These sampling sites were based on three land-use types, *i.e.*, farmland (10 sites, N1-10), lake (9 sites, D1-9), and ditches (6 sites, H1-6). A grab dredger was used to collect the surface sediments (0–10 cm) in ditches and lake area, and an original soil extractor was used to collect surface soil (0–10 cm) in farmland. Samples were placed in polyethylene bags

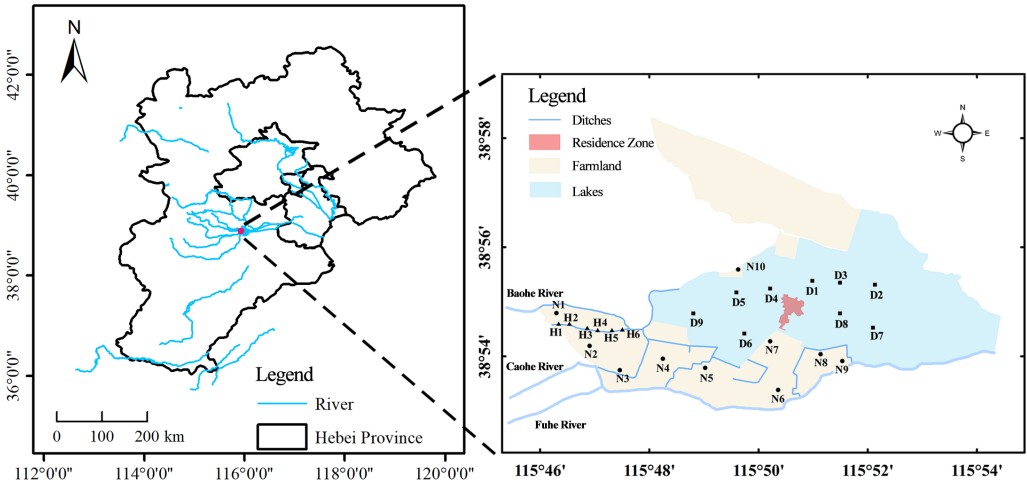

**Figure 1  Sampling site location.** N1–N10, farmland sampling point; H1–H6, ditch sampling point; D1–D9, lake sampling point.

and brought to the laboratory. After the samples were freeze-dried, the animal and plant residues and other remnants were removed. Pb, Zn, Cu, Cr, Cd, and Ni were determined using flame atomic absorption spectrophotometry (TAS-990 Super AFG) (*Ai et al., 2016*). Hg and As were determined using microwave digested-atomic fluorescence spectrometric (AFS-8520) (*Zhao et al., 2007*). Total phosphate was determined by the sulfuric acid and perchloric acid digestion method (*Huang et al., 2016*). Total nitrogen was determined by the micro-Kjeldahl method, and organic matter was determined by the potassium dichromate oxidation method (*Bao, 2000*). The pH was measured by portable pH analyzer (PHS-3C, INESA) using a 1:2.5 (*w/v*) suspension of solids in water. In the experiment process, the analytical work obeyed the quality control guideline. All samples were analyzed in triplicate and the deviation of duplicate samples was less than 5%.

## Evaluation method
### *Nemerow pollution index*
The Nemerow pollution index is used to evaluate nutrients in the sediments, which is convenient to compare the overall pollution degree of the target area with other regions (*Pandit et al., 2020*). This method considers the average and maximum value of the pollution index of a single element. The formula is based on *Li et al. (2021)*, as follows:

$$P_i = C_i / k_i \tag{1}$$

$$P_n = \left( \sqrt{\bar{P}_i^2 + [\max(P_i)]^2} \right) \Big/ 2 \tag{2}$$

where, $P_i$ is the single factor index, $C_i$ is the measured value, and $k_i$ is the standard value. The standard values of the two indicators (TN and TP) used in this study refer to the sediment evaluation guidelines issued by the Ministry of Environment and Energy of Ontario, Canada. Considering the lowest level degree of ecotoxicity effect in the evaluation

guidelines as the standard, the standard values of TN and TP were 0.55 and 0.60 g/kg, respectively (*Ye et al., 2019*). $\bar{P}_i$ is the average index of pollutants, $\max(P_i)$ is the maximum pollution index of a single pollutant. The pollution level classification is shown in Table S1.

### Geological accumulation index

This method uses the relationship between heavy metal content and geochemical background value to evaluate the degree of heavy metal pollution. The $I_{geo}$ is calculated using the following equation:

$$I_{geo} = \log2(C_n/kB_n) \tag{3}$$

where, $C_n$ is the measured content of the heavy metal, $B_n$ is the geological background value of the heavy metal. The background values of soils elements in Hebei Province were used in this study (*China Environmental Monitoring Station, 1990*) (*Xiang et al., 2020*). Therefore, the background values for As, Cd, Cu, Cr, Ni, Pb, Hg and Zn were 13.6, 0.094, 21.8, 68.3, 30.8, 21.5, 0.15, and 78.4 mg/kg, respectively. $k$ is the correction coefficient, which is determined according to the change of the background value caused by the rock difference in various places. In this case, this value is generally equal to 1.5 (*Zhuang et al., 2021*). Also, it is divided into seven levels according to the $I_{geo}$. The specific pollution degree is shown in Table S2.

### Potential ecological risk index

$RI$ is a comprehensive potential ecological hazard index. Compared with the $I_{geo}$, the $RI$ method considers the toxicity of the pollutants (*Lars, 1980*), gives the biological toxicity coefficients of eight pollutants, and divides the potential ecological hazard levels quantitatively (*Men et al., 2018*). The calculation formula is:

$$RI = \sum_{i=1}^{n} E_r^i = \sum_{i=1}^{n} T_r^i \times C_f^i \tag{4}$$

$$E_r^i = T_r^i \times C_f^i \tag{5}$$

$$C_f^i = C^i/C_n^i \tag{6}$$

where, $E_r^i$ is the potential ecological risk index of a single heavy metal, $T_r^i$ is the heavy metal biological toxicity response coefficient (As = 10, Cd = 30, Cr = 2, Cu = Ni = Pb = 5, Zn = 1, Hg = 40). $C_n^i$ is the background reference value of the heavy metal, and $C^i$ is the actual content of the heavy metal. The evaluation criteria for potential ecological hazards are shown in Table S3.

## Statistical analysis

The spatial distribution of the contaminant was mapped by geo-statistical method (Kriging interpolation, ArcGIS 10.4) (*Ganugapenta et al., 2018*). To test the effect of land-use types on pollutants, multivariate analysis of variance was conducted by IBM SPSS Statistics 24. Dual hierarchical clustering analysis is widely used in the analysis of environmental factors (*Li et al., 2015*). Dual hierarchical clustering analysis was performed using the heat map dendrogram tool of Origin 2018 to group the similar points and identify specific areas of

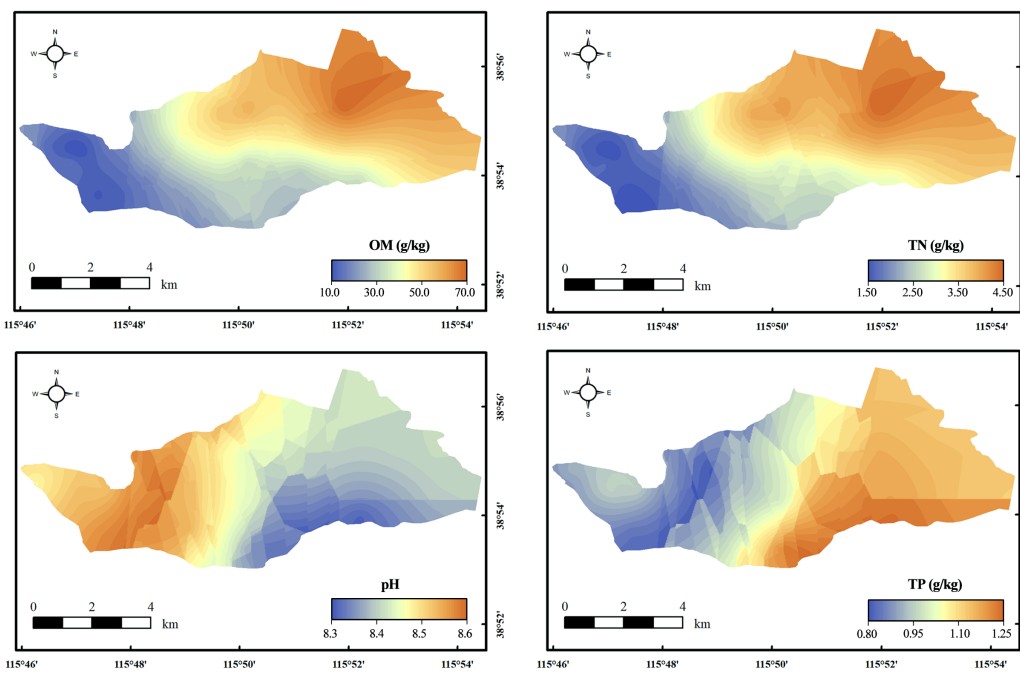

**Figure 2 Distribution of nutrients and pH.** The spatial distribution of the contaminant was mapped by Kriging interpolation.

contaminants. Principal component analysis was performed using R software to further analyze the relationship between contaminants (*Xu et al., 2014*). The script for principal component analysis is in Supplementary Files.

# RESULTS

## Spatial distribution of contaminants

The distribution of TN, TP, OM, and pH are presented in Fig. 2. OM and TN had similar spatial distribution characteristics, and the area with the higher TN and OM was located in the east of the study area, which corresponded to one land-use type: lake area. The contents of TN and OM in the lake area were $3.92 \pm 0.69$ g kg$^{-1}$ and $60.61 \pm 8.91$ g kg$^{-1}$ respectively (Table 1). The average content of TN and OM was in the following order: lake > farmland > ditch. On the contrary, the higher pH was in the west, corresponding to the farmland area. The area with the higher TP content was located in the south of the study area, close to a village. The average content of TP in lake, farmland, and ditches were 0.93 g kg$^{-1}$, 0.89 g kg$^{-1}$, 0.91 g kg$^{-1}$, respectively.

There was a regional difference in heavy metals (Fig. 3). The area with higher heavy metal content was the farmland. The heavy metals Cr and Ni had similar distributions and the content gradually decreased from west to east. The content of Cu, Zn, As, and Hg were generally higher in the south and lower in the north. These four pollutants were mainly concentrated in farmland areas. Pb and Cd were mainly concentrated in the middle of the study area, and their contents diffused to the surroundings and decreased. The results of

**Table 1 Content of pollutants in different land-use types.**

| | Background values | Land type (mean ± standard deviation) | | |
|---|---|---|---|---|
| | | Farmland (*n* = 10) | Ditch (*n* = 6) | Lake area (*n* = 9) |
| Pb (mg/kg) | 21.5 | 49.25 ± 9.94 | 41.70 ± 6.90 | 42.98 ± 3.14 |
| Cd (mg/kg) | 0.094 | 0.10 ± 0.03 | 0.04 ± 0.04 | 0.09 ± 0.03 |
| Zn (mg/kg) | 78.4 | 96.20 ± 16.44 | 73.50 ± 21.34 | 88.78 ± 5.83 |
| Cu (mg/kg) | 21.8 | 42.20 ± 8.68 | 33.50 ± 13.29 | 36.78 ± 3.11 |
| Cr (mg/kg) | 68.3 | 53.50 ± 13.54 | 49.00 ± 5.55 | 46.11 ± 2.71 |
| Ni (mg/kg) | 30.8 | 52.00 ± 4.99 | 47.67 ± 13.22 | 47.89 ± 2.57 |
| As (mg/kg) | 13.6 | 11.04 ± 3.17 | 8.02 ± 2.78 | 8.74 ± 0.84 |
| Hg (mg/kg) | 0.15 | 0.06 ± 0.03 | 0.02 ± 0.01 | 0.03 ± 0.03 |
| TN (g/kg) | – | 1.97 ± 0.48 | 1.60 ± 0.91 | 3.92 ± 0.69 |
| TP (g/kg) | – | 0.89 ± 0.19 | 0.91 ± 0.17 | 0.93 ± 0.10 |
| OM (g/kg) | – | 30.86 ± 8.45 | 23.48 ± 13.72 | 60.61 ± 8.91 |

multivariate analysis of variance showed that land-use type had a significant impact on the distribution of pollutants ($p < 0.05$) (Table S4). The average contents of all heavy metals and As in farmland soil were higher than those in lake and ditch sediments (Table 1). The heavy metals Pb, Cu, and Ni exceeded the background value in three land-use types. The content of Zn in farmland and lake area was higher, which exceeded the background value by 1.22 and 1.13 times, respectively. The Cd in the farmland exceeded the background value by 1.06 times. Moreover, the content of Cr, As, and Hg were lower than the background value in all regions.

## Evaluation of total nitrogen and total phosphorus

The pollution index of TN and TP at each sampling point of the lake and ditch area was greater than 1 (Table S5), indicating that there were nitrogen and phosphorus pollution ($1.32 < P_{TN} < 8.69$, $1.18 < P_{TP} < 2.35$). Nitrogen pollution of sediments in the study area showed spatial differences. The $P_{TN}$ ranges of ditches and lake areas were 1.32–5.56 and 2.91–8.69, respectively. All points in the lake area were heavily polluted, except for D9. In fact, the degree of TP pollution varied among different regions, except for two sampling points, N6 and D8, which were moderately polluted, and other points were slightly polluted. The comprehensive pollution index showed that more than 90% of the lake area was seriously polluted by nitrogen and phosphorus.

## Pollution evaluation and risk assessment of heavy metals and As

The $I_{geo}$ values of all the analyzed metals (Pb > Cu > Ni > Zn > Cd > As > Cr > Hg) were 0.49, 0.28, 0.18, −0.45, −0.99, −1.06, −1.17, and −3.03, respectively (Fig. 4). Among them, the Pb, Cu, and Ni pollution degree was "low polluted" showing an $I_{geo}$ range of 0 to 1. The $I_{geo}$ of other heavy metals was less than 0, indicating that their pollution degree was "unpolluted". The metal pollution levels of the three land types were consistent with the overall pollution levels. Fig. 4D shows that the range of $I_{geo}$ in the lake area is very small,

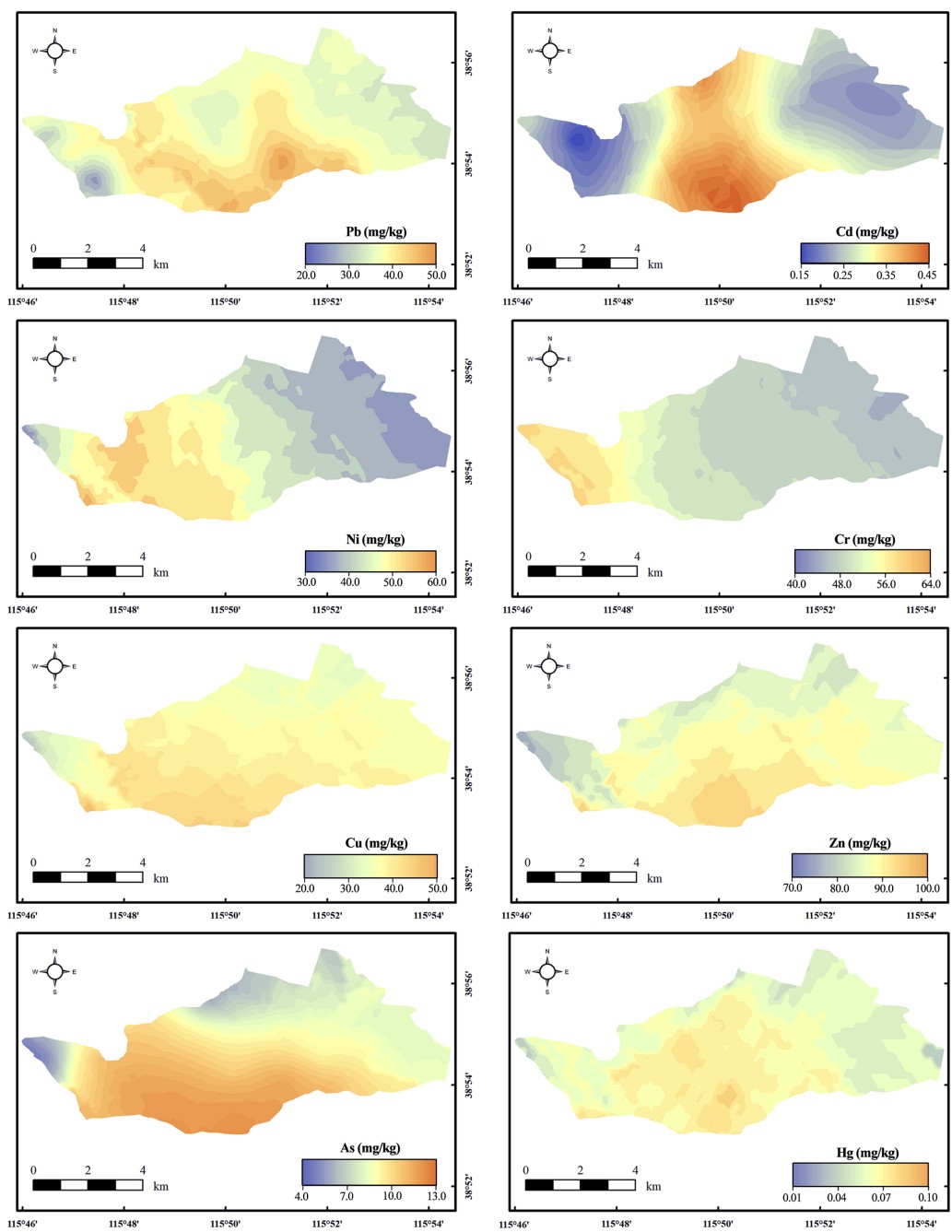

**Figure 3 Distribution of heavy metals and As.** The spatial distribution of the contaminant was mapped by Kriging interpolation.

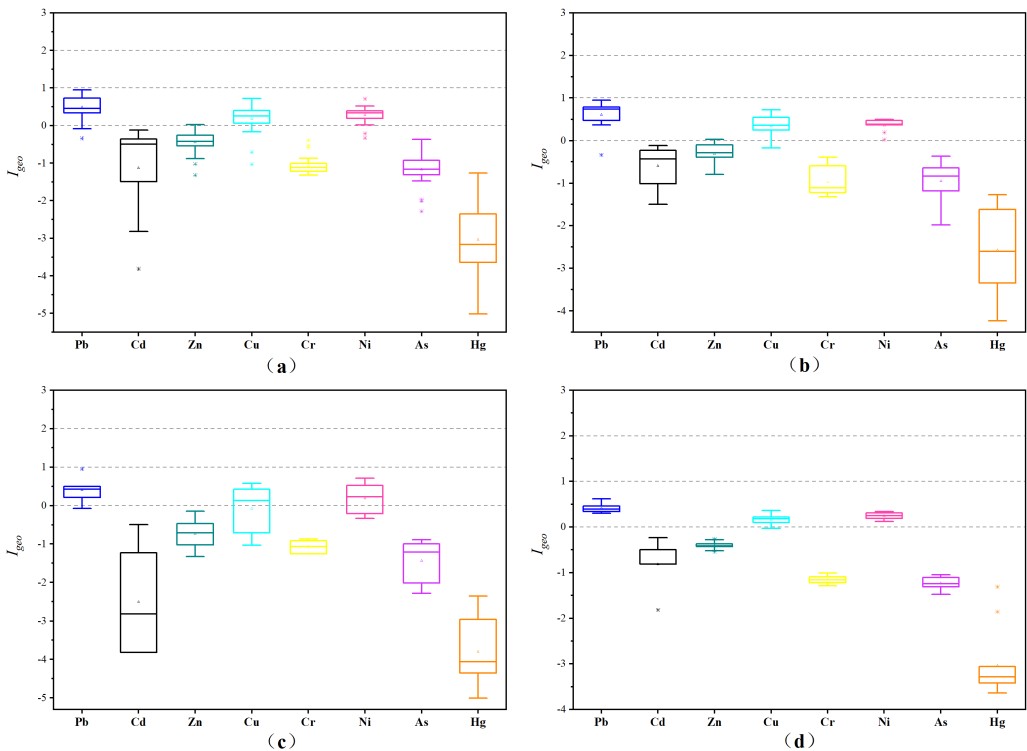

**Figure 4** $I_{geo}$ **in sediments of different regions.** (A) Whole study area; (B) farmland; (C) ditch; (D) lake area.

indicating that the sources or accumulation modes of heavy metals at different lake points are similar. The wide range of $I_{geo}$ in farmland and ditches indicates the accumulation of heavy metals in farmland and ditches is affected by multiple pollution sources.

The order of the $E_r^i$ values was Cd > Pb > Cu > Ni > As > Hg > Cr > Zn (Fig. 5), and the $E_r^i$ of the eight elements were all lower than 40, indicating low ecological risks. The dominant factor affecting $RI$ was Cd, which represents the possible ecological risk to the biological community. The $E_r^i$ values of Cd in farmland, ditch, and lake were 31.28, 14.36, 27.30, respectively, which were different and indicated a higher potential ecological risk of Cd in farmland. The $RI$ values of the three land-use types in the study area were 84.21, 71.34, 50.78 for farmland >lake >ditch, respectively. Therefore, the potential ecological risk of heavy metals in the farmland was higher than that in other regions. Comprehensive analysis showed that the rankings of the $I_{geo}$ and $E_r^i$ were different. Moreover, both indicated that Pb, Cu and Ni ranked higher and represented the main heavy metal risk factors.

## Mutivariate statisistical analysis

Dual hierarchical clustering analysis showed that all elements were divided into two major categories (Fig. 6). The first category was TN, TP, OM and the second category was As and heavy metals. The indicators were further clustered and divided into four categories. Cr was a single cluster, and the main pollution sites were N1, N2, and N3. In addition, Cd was located in a single cluster, and the concentration at N5, N6, N7, and N10 was

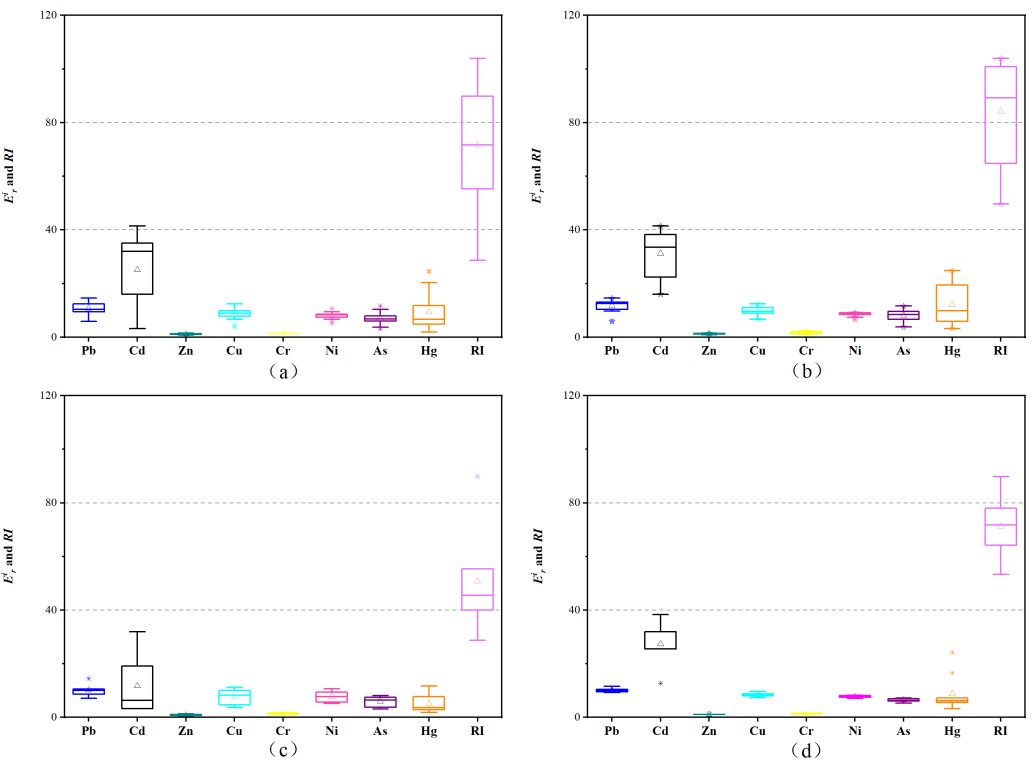

**Figure 5** **The $E_r^i$ and $RI$ of different regions.** (A) Whole study area; (B) farmland; (C) ditch; (D) lake area.

significantly higher than compared to other sites. Pb, Zn, Cu, Ni, As, and Hg were classified into one category, indicating that the distribution of these elements was similar. Moreover, TN, OM, and TP were classified into one category. Furthermore, a horizontal dendrogram clustering was performed, resulting in five clusters. The farmland points were mainly found in the fifth cluster, lake points were in the fourth cluster, and ditch points were in the first, third, fourth, and fifth clusters. The clustering arrangement agreed to the actual pollution distribution of study area. Heavy metals and As were high in farmland soils, while nutrients were high in lake sediments.

The principal component analysis showed that the eigenvalues of the four principal components greater than 1 explained 84.2% of the total variance (Table S6). The PC1 explained the variance of 39.8%, which included Cu, Ni, As, Pb, and Zn. The PC2 accounted for 22.1% of the total variance, which was related to TN and OM. Finally, PC3 and PC4 were dominated by Cr and Cd, respectively. The relationship between pollutants is consistent with the results of Pearson correlation analysis (Fig. S1). Figure 7 shows the contribution of elements to the sampling point. The sampling points in the lake area were mainly found in the first and second quadrants, TN and OM were the main pollutants, and Cd, Zn, and TP also contributed to the first quadrant. Moreover, the farmland sampling points were in the fourth quadrant, and these sampling points were mainly affected by Ni, As, Cu, Cr, Pb, and Zn. Additionally, the points of the ditches were relatively scattered,

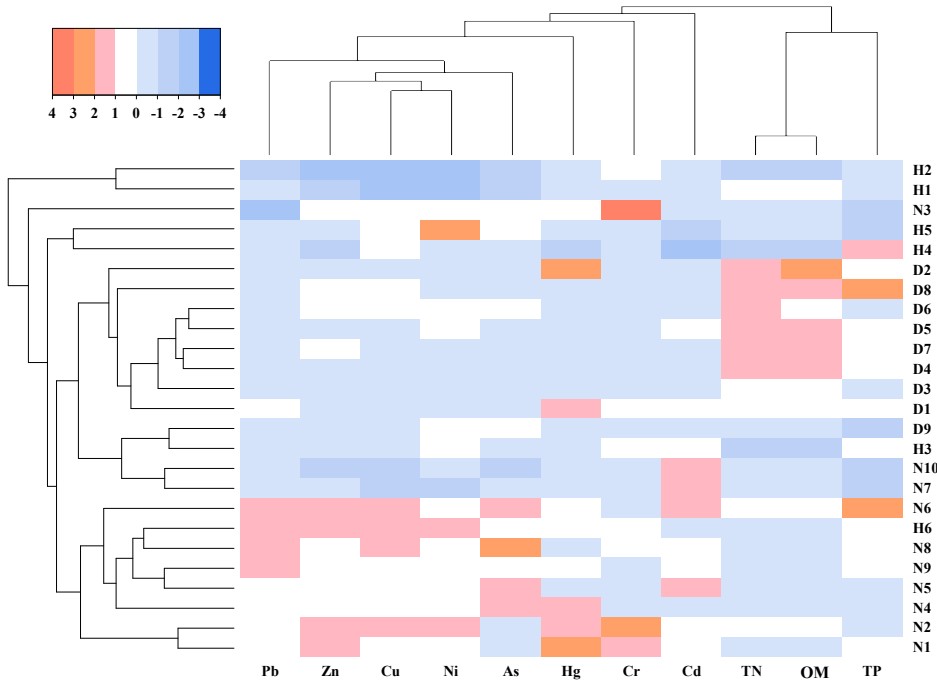

**Figure 6  Heatmap of clustering heavy metals, As, and nutrients.**

probably due to the surrounding environment which can have a great impact on these points.

## DISCUSSION

### Pollution source identification

Dual hierarchical clustering analysis and principal component analysis showed that Cu, Ni, As, Pb, and Zn had similar sources in the study area. Specifically, the average values of Cu, Ni, Pb, and Zn exceed the background value, except for As. In fact, these contaminants had the highest contents in farmland soils. Previous studies have shown that the raw material for producing phosphate fertilizers contains As (*Gupta et al., 2014*). And in the production process of phosphate fertilizer, heavy metals may also be brought into the fertilizer (*Zhang et al., 2016*). Therefore, the long-term use of phosphorus fertilizer may be the main cause for the high content of heavy metals and As in farmlands. In addition, some pesticides such as organic arsenic fungicides and lead arsenate pesticides contain heavy metals. Consequently, the use of pesticides can also contribute to pollution by heavy metals (*Fallah et al., 2021*). Therefore, Cu, Ni, As, Pb, and Zn may be identified as an agricultural source. Cr is a group of its own. The Cr content in the three land-use types was lower than the background value, which indicated that the sediments and soils were not polluted by Cr. These results suggest that Cr could mainly come from a geological but not an anthropogenic source. The analysis result of Cd shows that it has different pollution sources from other elements. Previous research has shown that the main sources of Cd in soils are mining, industry, and automobile pollution (*Das, Samantaray & Rout, 1997*).
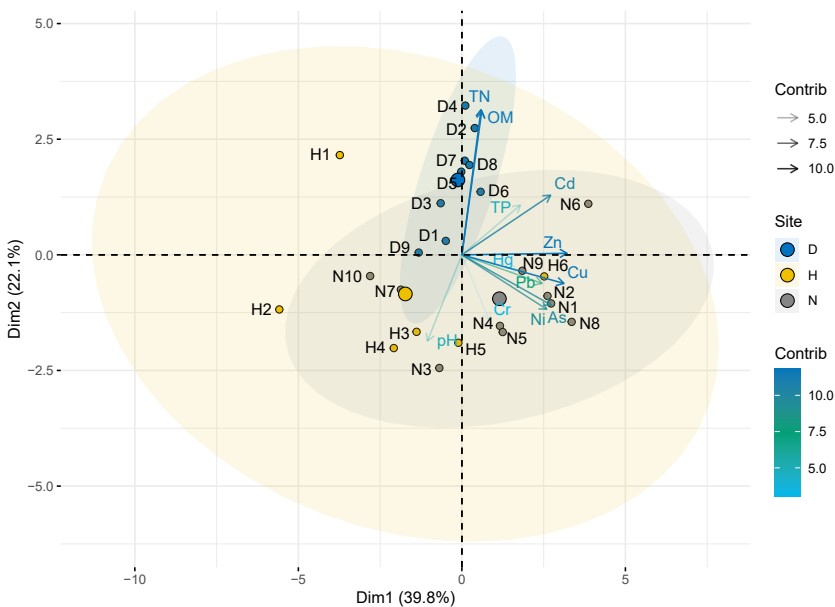

**Figure 7** Loading plot of the principal component analysis.

Previously, there were many companies involved in papermaking, electric power, printing, and dyeing in the upstream of the study area. In fact, industrial wastewater was discharged into the Fuhe River and then flowed into wetland (*Hu et al., 2011*). Thus, Cd may come from industrial sources.

The sources of TN and OM were similar. The content of TN and OM in lake sediments was higher than that of farmland and ditches. In fact, dead organisms such as plants, algae, and mollusks in the lake can accumulate in the benthic zone of the lake, merging with the clay silt (*Xing et al., 2021*). Also, the poor water flow of Zaozhadian Lake is more likely to cause a reduction environment at the bottom of the lake, promoting the accumulation of TN and OM in sediments. Therefore, TN and OM in sediments mainly come from animal and plant residues. Studies have shown that one of the main factors for the changes in OM and TN content in the wetland could be the types of vegetation (*Zhao et al., 2020*). Also, point source pollution will directly affect the distribution of pollutants (*Yang et al., 2018*). As mentioned above, the nearby villages in the middle of the study area had the highest TP content. In this area, most of the poultry was raised on the nearby water surface, so the feces were directly discharged into the lake. Moreover, the villagers directly discharged domestic sewage into the water area, which also resulted in higher TP content.

## The impact of land-use type changes on the pollutants

The TN, TP, and OM in farmland are lower than other land-use types, which means that the soil nutrients decreased after the lake area was converted to farmland. Firstly, the accumulation of nutrients in soil decreases due to the reduction of aquatic animals and plant decay. Secondly, crops need to absorb nutrients in the growth process, and long-term planting of crops consumes nutrients in the soil. Finally, long-term traditional farming

destroyed soil aggregates and accelerated the loss of soil organic matter (*Chambers et al., 2014*).

The data presented in this study and in *Zheng et al. (2016)* showed that heavy metals content increased after the lake area was transformed into cropped land. The conversion of lake area to farmland increased the way for heavy metals to enter the wetlands. Cultivation typically increases the input of agrochemicals in the form of fertilizers, pesticides, and manures, which has led to the enrichment of some heavy metals in soil (*Wang & Xu, 2015*). However, many studies also showed that soil heavy metals decreased after lake was converted to farmland (*Bai et al., 2010*). The reason for these variances is due to the difference in soil properties and cultivation practices that alter the geochemical behavior of heavy metals in soils (*Sun et al., 2019*; *Jiao et al., 2014*). Moreover, the conversion of land-use type could also change the physical and chemical properties of soil. For instance, it has been widely reported that the soil organic matter has a large capacity for sorption of heavy metals in the soils (*Yan et al., 2017*). In this study, the content of organic matter in sediments decreased from $60.61 \pm 8.91$ g kg$^{-1}$ to $30.86 \pm 8.45$ g kg$^{-1}$ after the lake was transformed into farmland. Therefore, the consumption of organic matter reduced the ability of soil to retain heavy metals, which may result in an increase of heavy metal mobility and phytotoxicity in study area.

### Implications for land-use planning and water quality maintenance

In farmlands, most of the heavy metals in the soil have originated mostly from man-made sources than natural sources. After the lake was transformed into farmland, the accumulation of heavy metals in wetland system accelerated. Heavy metals are overloaded in soils cause a severe environmental risk. In addition, heavy metals from farmland soils can be released to lakes through agricultural runoff and affect aquatic communities (*Fernández et al., 2005*). For lakeshore wetlands, these effects are more obvious because the lake water is stagnant, allowing heavy metals to easily accumulate (*Sun et al., 2019*). A greater area of farmland can result in a high risk to the regional environmental system. Relevant departments should take effective measures to reduce the toxic risks of contaminants through exogenous control and to adjust the land pattern of the Zaozhadian Lake.

The results of this study provide useful information that can be used to emphasize on potential long-term impacts of land-use type change to mitigate or minimize negative impacts. However, this study was unable to demonstrate any relationship between changes of soil properties and the migration of pollutants. Changes in water content, exposure to the air, crops grown and methods of fertilization create a complex scenario that influences pollutant migration, which should be further studied. Moreover, we suggest that the adjustment of wetland land-use types should be based on clarifying the factors affecting the migration and transformation of pollutants and changes in toxicity.

## CONCLUSIONS

Spatially, there were differences in pollutant content under different land-use types. The sediments in the lake area have high TN and OM content, and the farmland soils have a higher content of heavy metals. Also, Zaozhadian Lake soils can be described as low

polluted by heavy metals. Among the selected heavy metals, Pb shows the highest degree of pollution and Cd shows the highest ecological risk. Moreover, the pollution level of heavy metals in the farmland area was lightly polluted, and most of the ditches and lake areas were not polluted. Furthermore, Cu, As, Ni, Pb, and Zn were primarily derived from agriculture pollution (*e.g.*, phosphate fertilizer and pesticides). In contrast, Cd and Cr mainly originated from industrial and natural sources, respectively. The nutrients were generally controlled by the decomposition of aquatic animal and plant and human activities. Land use changes could affect some contaminants concentrations in wetland soils. The change of lake to farmland decreased the concentrations of TN and OM, and increased the concentrations of heavy metals and As. Relevant departments should take effective measures to reduce the toxic risks of contaminants through exogenous control and to adjust the land pattern of the Zaozhadian Lake. Additionally, the effects of changes in soil properties on pollutant migration and toxicity should be further studied.

## ACKNOWLEDGEMENTS

We thank our colleagues and research staff members for their constructive advice and help.

### Funding

This research was supported by the Major Science Technology Program for Water Pollution Control and Treatment of China (2018ZX07110-006), the National Natural Science Foundation of China (NO. 51778054), the Advanced Talents Incubation Program of Hebei University (521000981379, 521000981281 and 801260201305), and the Innovation funding project for Postgraduates of Hebei University (hbu2020bs002). The funders had no role in study design, data collection and analysis, decision to publish, or preparation of the manuscript.

### Grant Disclosures

The following grant information was disclosed by the authors:
Major Science Technology Program for Water Pollution Control and Treatment of China: 2018ZX07110-006.
National Natural Science Foundation of China: 51778054.
Advanced Talents Incubation Program of Hebei University: 521000981379, 521000981281, 801260201305.
Innovation funding project for Postgraduates of Hebei University: hbu2020bs002.

### Competing Interests

The authors declare there are no competing interests.

### Author Contributions

- Yangyang Han performed the experiments, analyzed the data, prepared figures and/or tables, and approved the final draft.

- Hongjie Wang, Guangming Zhang and Ling Liu conceived and designed the experiments, authored or reviewed drafts of the paper, and approved the final draft.
- Shengqi Zhang performed the experiments, prepared figures and/or tables, and approved the final draft.
- Xingchun Liu conceived and designed the experiments, prepared figures and/or tables, and approved the final draft.

### Field Study Permissions

The following information was supplied relating to field study approvals (i.e., approving body and any reference numbers):

Sample collecting and field experiments were approved by the Major Science Technology Program for Water Pollution Control and Treatment of China (2018ZX07110)

### Data Availability

The raw measurements and the R script are available in the Supplementary Files.

### Supplemental Information

Supplemental information for this article can be found online at http://dx.doi.org/10.7717/peerj.12885#supplemental-information.

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
