# Peer review of "Distribution, ecological risk assessment and source identification of pollutants in soils of different land-use types in degraded wetlands"

_PeerJ, doi:10.7717/peerj.12885_

## Round 0.1 · original submission · Major Revisions

I have now heard back from three reviewers who provided constructive comments on your manuscript. All of them were unanimous in not only recommending major revision, but also emphasizing that the English language needs to be heavily worked on throughout the text. I concur with them, it's quite hard to understand the authors' message in some paragraphs. So in addition to responding to the reviewers' critique, you will need to work out the English language and writing style.

However, the most pressing problem with the manuscript is by far the data analysis. Authors simply didn't provide enough details to allow reproducibility, simply mentioning they conducted "correlation analysis, PCA" etc. This is not acceptable. You need to provide enough details as to allow anyone to reproduce the analysis. And since you conducted data analysis, at least in part, in R, I highly recommend you to share the R script and data.

Also, you need to use a MANOVA to test the effect of land-use types on all pollutants at once, not many ANOVAs for each substance. This multiple testing is incorrect. Additionally, since you already have a PCA, you don't need a pair-wise correlation, so delete this analysis
Avoid using too many acronyms, there are already lots of them throughout the text, which gives it a telegraphic style.

Most importantly, large parts of the results section contain discussion of the results. This also needs to be separated accordingly.
I'm only willing to continue to consider your article if these central issues are solved.

Also, notice PeerJ uses a structured abstract. I highly recommend you adhere to it.

I have made several comments and edits directly in the pdf attached. Please, include a response to them in your rebuttal letter

Reviewer 1 ·

Basic reporting

(1) A proof reading by a native English speaker should be carefully conducted to improve both language and organization quality.
There should be a space between values and units. Lines 26-27: “3.71 ± 1.03 g/kg”, “57.17 ± 15.16 g/kg”.
The “Figure” and “Fig” are improperly mixed.
There is an inappropriate use of italics. Line 68: “Nemerow pollution index (NPI)”.
Line 105: The “&” should be changed to “and”.
(2) Line 126: (GB15618-1995, GB/T17141-1997, HJ491-2019). Which national standard is this?
(3) Line 138: Table S1 is not found in the manuscript.
(4) Lines 143, 155 and 161: No spaces should be left before “where”.
(5) Line 172: Distribution of nitrogen, phosphorus. However, only TN is mentioned in the content.
(6) Lines 173 and 174: It can be seen….. What does "it" stand for here?
(7) Line 390: it was found that….. What does "it" stand for here?
(8) Line 411: We know land use….. The author avoids using the first person.
(9) There are few references in recent 3 years.
(10) The format of references in the paper can not meet the requirements of journals.

Experimental design

The manuscript lacks the importance analysis of the research area.

Validity of the findings

There is a lack of in-depth discussion in the manuscript.

Additional comments

The author needs to examine the whole manuscript in detail.

Reviewer 2 ·

Basic reporting

1. This manuscript needs careful editing by someone with expertise in technical English editing, paying particular attention to English grammar and sentence structure so that the goals and results of the study are clear to the reader.
2. This study is mainly focusing on the pollutants of different land-use types. Therefore, your introduction needs more detail about how different land-use types influence the contents and distribution of pollutants in the previous study.
3. The paper is overall well-structured, while the results need to be clearly and simply stated without discussing how they related to the previous research in this part.

Experimental design

It has described clearly enough the methods to allow others to repeat them, and evaluation methods are also appropriate ways to assess the status of pollution. However, both RI and PLI could identify the comprehensive pollution level of heavy metals. I think it would be best to choose one of them.

Validity of the findings

I suggest the authors state the conclusions as clearly as possible.

Additional comments

The current study investigated the distribution and sources of heavy metals, As, and nutrients in a degraded wetland by measurement methods and multiple evaluation methods. It is a topic of interest to the researchers in the related areas but the paper needs very significant improvement before acceptance for publication.

Reviewer 3 ·

Basic reporting

The writing of the article is relatively standardized, but the logic of English expression is not enough. It is recommended to further improve and polish. “Introduction”: author introduced the role of sediments and soils on the "sink" of pollutants and the importance of evaluation, but does not introduce the background of the impact of land-use types changes on pollutants. It is recommended to supplement and straighten out the logic. In addition, the GIS-based processing methods and pollutant evaluation indicators (Line64-84) introduced in the introduction have been widely used, and there is no need to repeat them. If this content is needed, it is recommended to add to the "Materials and Methods" section. The introduction of Line84-94 about study area is recommended to be placed in the "Study area and sample collection" section of the “Materials & Methods” section. Moreover, it is far-fetched to put forward the research question, recommended to reorganize and write.

Experimental design

The experimental design is reasonable, and recommended to supplement the specific depth of surface sediment collection (Line119).

Validity of the findings

The conclusion can basically summarize the results, and recommended to supplement future research prospects.

Additional comments

1) Some evaluation indicators are used in the article to evaluate the degree of pollution, and source analysis is carried out, so the title of the article can be appropriately corrected to match the content of the article
2) It is recommended to add the significance of the study in the abstract
3) Line111, it is recommended to supplement the relevant information of three rivers, which can correspond to the information of Line258. It would be better if the rivers can be marked in Figure 1.
4) Line174-175 and Line195-196 are duplicated, which leads to confusion in the logic of this paragraph. It is recommended to rearrange the language
5) Line180: Is this water area? Or Lake area?
6) Line201,What kind of change pattern?
7) 7) It is recommended to directly use p < or > 0.01 or 0.05 to express the significance in the text to be more concise and clear.
8) Line221, There is a problem with the english expression.
9) Line249 "The pollution degree was clean" has a problem, it is better to change to "the pollution degree was lower".
10) The references in the text and the reference list do not correspond.
11) Line352-358 language logic needs to be reorganized.
12) Line366-367, Line372-373 add corresponding references.

---

## Round 0.2 · Minor Revisions

Thank you for preparing a much improved version of the manuscript with a detailed response to the reviewers’ comments and mine. However, there are still some space for improvement. See the last minor comments by R2.

Reviewer 1 ·

Basic reporting

The author has revised the relevant content.

Experimental design

The author has revised the relevant content.

Validity of the findings

The author has revised the relevant content.

Reviewer 2 ·

Basic reporting

no comment

Experimental design

no comment

Validity of the findings

no comment

Additional comments

Title:” Distribution, ecological risk assessment and source identification of pollutants in soils of different land-use types in degraded wetlands”

I have read the revised MS and I think it is the scope of Peer J, but I am afraid a minor revision is needed for the authors to improve the MS before acceptance for publication.

Specific comments:

1. PLZ add a conclusion or suggestion sentence in the Results portion in the Background part.
2. line 46-48, PLZ add a conclusion or suggestion sentence in the Results portion.
3. line 26, a specific experiment type should be mentioned here.
4. line 56, sorption is also an existed form for heavy metals in the soil environment.
5. line 57-58, revise this sentence, what is ‘the accumulation of these pollutants in wetlands’?
6. line 59, ‘soils pollution’ is a mistake. And line 294, ‘soils heavy metals’, check throughout this paper.
7. line 73, differences in soil pollutants? Types? concentration? Distribution?
8. line 84, what is the ‘to data’? PLZ revise the sentence.
9. line 87, what is the ‘to our knowledge’? PLZ revise the sentence.
10. line 173, revise the unit format.
11. Authors should further improve the writing. More attention should be paid to the logical improvement.
12. The Figures were finished well.
13. Land pattern does determine much of the biogeochemical processes including the changes of soil pH, microorganism communities, the availability of the heavy metals, etc. Authors should further focus on the mechanism for the reason why the land pattern can efficiently control the distribution, risk assessment and the bioavailability of the heavy metals in the soils in the wetland system.

---

## Round 0.3 · accepted · Accept

I believe the authors have done a good job and addressed nicely all the comments raised by the two reviewers and me. I’m glad to move your manuscript forward to production. Congratulations